*The Company of*
**Biologists**

## RESEARCH ARTICLE

# Generative model for the first cell fate bifurcation in mammalian development

Maria Avdeeva[1,*,‡], Madeleine Chalifoux[2,3,*], Bradley Joyce[3], Stanislav Y. Shvartsman[1,2,3] and Eszter Posfai[3,‡]

## ABSTRACT

The first cell fate bifurcation in mammalian development directs cells toward either the trophectoderm (TE) or inner cell mass (ICM) compartments in pre-implantation embryos. This decision is regulated by the subcellular localization of a transcriptional co-activator YAP and takes place over several progressively asynchronous cleavage divisions. As a result of this asynchrony and variable arrangement of blastomeres, reconstructing the dynamics of the TE/ICM cell specification from fixed embryos is extremely challenging. To address this, we developed a live-imaging approach and applied it to measure pairwise dynamics of nuclear YAP and its direct target genes, CDX2 and SOX2, which are key transcription factors of the TE and ICM, respectively. Using these datasets, we constructed a generative model of the first cell fate bifurcation, which reveals the time-dependent statistics of the TE and ICM cell allocation. In addition to making testable predictions for the joint dynamics of the full YAP/CDX2/SOX2 motif, the model revealed the stochastic nature of the induction timing of the key cell fate determinants and identified the features of YAP dynamics that are necessary or sufficient for this induction. Notably, temporal heterogeneity was particularly prominent for SOX2 expression among ICM cells. As heterogeneities within the ICM have been linked to the initiation of the second cell fate decision in the embryo, understanding the origins of this variability is of key significance. The presented approach reveals the dynamics of the first cell fate choice and lays the groundwork for dissecting the next cell fate decisions in mouse development.

KEY WORDS: Mouse, Preimplantation, First cell fate decision, Live imaging, Bayesian modeling

## INTRODUCTION

Early development of most well-studied model organisms follows a largely deterministic developmental pattern where differentiation can be attributed to pre-existing asymmetries derived from sperm entry and/or nonuniformly distributed maternal factors (Davidson, 1990). In the mouse, the leading model for studying mammalian development, such early asymmetries are not major factors

contributing to differentiation (Zhu and Zernicka-Goetz, 2020; Zhang and Hiiragi, 2018). Instead, differentiation, which is first detected after the fourth cleavage that generates the 16-cell morula, is driven by self-organization (Cockburn and Rossant, 2010). Because mouse embryos do not follow an invariant cleavage pattern, the morulae vary in shapes and arrangements of blastomeres (Dard et al., 2009; Watanabe et al., 2014; Korotkevich et al., 2017; Niwayama et al., 2019; Fabrèges et al., 2024). This geometric variability has been shown to be an essential component of differentiation, with geometric cues driving cells towards their respective fates (Royer et al., 2020; Johnson and Ziomek, 1981; Tarkowski and Wróblewska, 1967). Furthermore, spatial and temporal variability in cleavage patterns and resulting gene expression variability have been proposed to be a driver of robustness in the first cell fate decisions (Fabrèges et al., 2024; Niwayama et al., 2019; Holmes et al., 2017; Saiz et al., 2020). As a consequence of this unique mode of development, quantitative models of the early mouse embryo must be distinct from the essentially deterministic models that suffice for embryos that depend on the invariant pattern of cleavages and pre-existing asymmetries.

Here, we establish such a model for the first cell fate bifurcation in the mouse embryo, as cells are directed to either the inner cell mass (ICM), which is the precursor of the fetus and certain extra-embryonic tissues, or the trophectoderm (TE), which is the precursor of the placental cell types (Fig. 1A). The specification of these two cell types is associated with the expression of well-known lineage-specific transcription factors (TFs), such as SOX2 (ICM) and CDX2 (TE) (Wicklow et al., 2014; Strumpf et al., 2005). Both CDX2 and SOX2 are regulated by the Hippo signaling pathway and its downstream effector, the Yes-associated protein (YAP), where YAP serves as an activator of *Cdx2* and a repressor of *Sox2* expression (Nishioka et al., 2009; Wicklow et al., 2014; Frum et al., 2018, 2019) (Fig. 1B). YAP activity is determined by its subcellular localization, which in turn is regulated by apicobasal polarity of cells, with polarized cells localizing YAP to the nucleus and apolar cells sequestering YAP in the cytoplasm (Nishioka et al., 2009; Hirate et al., 2013; Leung and Zernicka-Goetz, 2013).

Segregation of ICM and TE fates largely takes place between the 8- and 32-cell stages. At the 8-cell stage, gene expression variability among cells is low, and individual cells display bipotency towards ICM and TE fates (Posfai et al., 2017). By the 32-cell stage, TE cells line the outside of the embryo and ICM cells are located on the inside. This sequence of events was largely deduced from images of fixed embryos, which offer only snapshots of nuclear YAP and its transcriptional effects, with limited live imaging datasets of individual components available to date (McDole and Zheng, 2012; Toyooka et al., 2016; Gu et al., 2022). Using these datasets for quantitative understanding and modeling of the underlying dynamics is a highly challenging task (Krupinski et al., 2011; Holmes et al., 2017; De Caluwé et al., 2019; Nissen et al., 2017; Cang et al., 2021; Ramirez Sierra and Sokolowski, 2024). Using live imaging together with computational cell segmentation and tracking, we

[1]Center for Computational Biology, Flatiron Institute, Simons Foundation, New York, NY 10010, USA. [2]Lewis-Sigler Institute for Integrative Genomics, Princeton University, Princeton, NJ 08544, USA. [3]Department of Molecular Biology, Princeton, NJ 08544, USA.
*These authors contributed equally to this work

‡Authors for correspondence (mavdeeva@flatironinstitute.org; eposfai@princeton.edu)

M.A., 0000-0002-6366-2269; E.P., 0000-0002-4871-6652

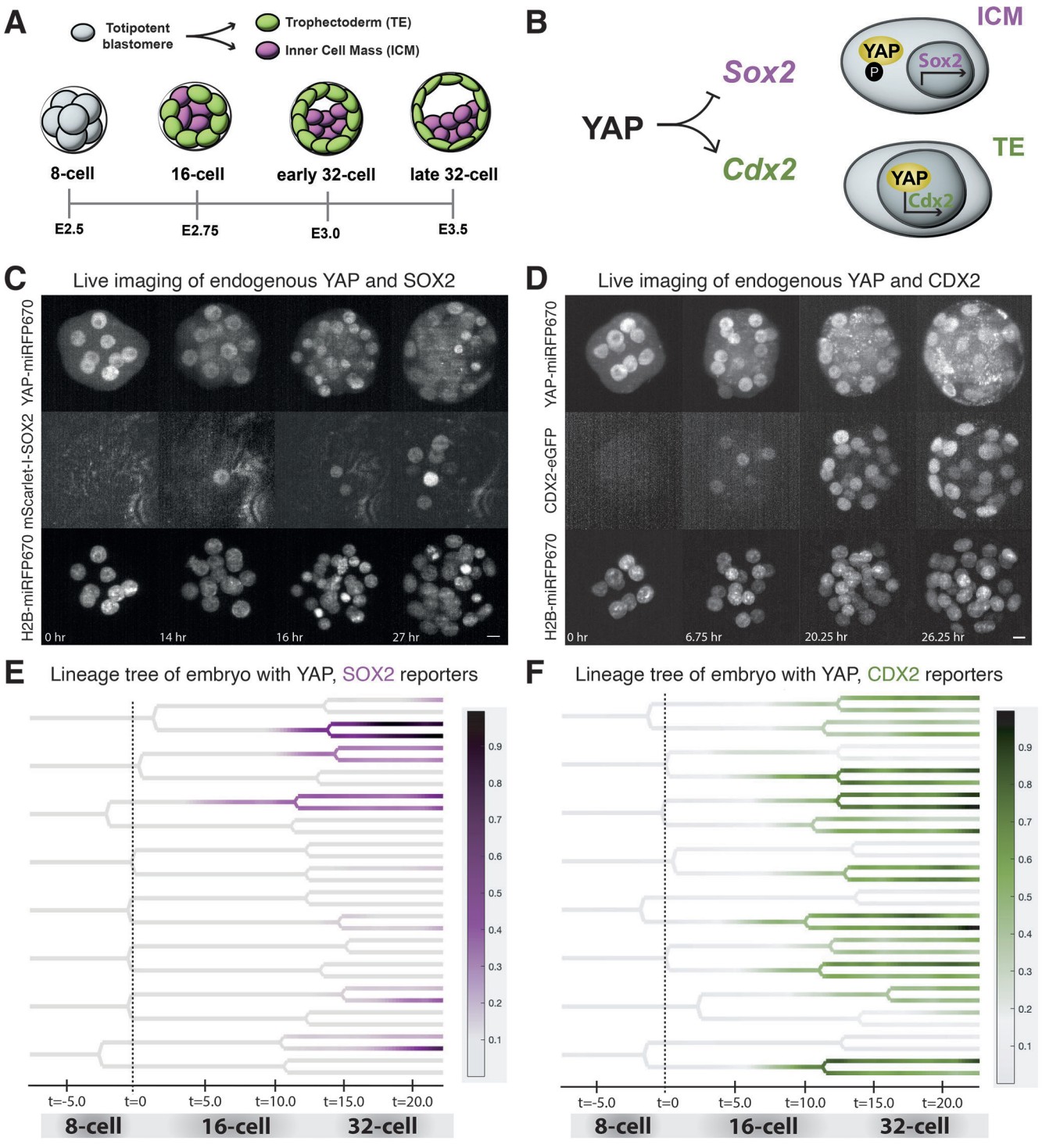

**Fig. 1. Live imaging of endogenously tagged YAP-miRFP670, mScarlet-I-SOX2 and CDX2-eGFP during the first fate decision in preimplantation embryos.** (A) A developmental timeline of trophectoderm (TE) and inner cell mass (ICM) specification. The first cell differentiation event, between the TE (green) and ICM (purple), initiates at the 16-cell stage, establishing distinct inner, ICM, and outer, TE, cell populations in the 32-cell stage embryo. Embryos in this study are imaged until the 32-cell stage. Days post-fertilization are denoted by embryonic day (E). (B) *Sox2* and *Cdx2* expression is regulated by YAP. In ICM cells, YAP is phosphorylated and retained in the cytoplasm, facilitating *Sox2* expression. In TE cells, YAP is predominantly localized to the nucleus where it activates expression of *Cdx2*. (C) Time-lapse images of a representative embryo expressing YAP-miRFP670, mScarlet-I-SOX2 and H2B-miRFP670 from the 8-cell through the late 32-cell stage. A *z*-stack was acquired every 15 min for YAP and H2B channels, and every 30 min for the SOX2 channel. Maximum intensity projections (MIPs) are shown. Unit of time is hours. Scale bar: 10 µm. (D) Time-lapse images of a representative embryo expressing YAP-miRFP670, CDX2-eGFP and H2B-miRFP670 from the 8-cell stage through the late 32-cell stage. Scale bar: 10 µm. (E) Lineage trees from an embryo expressing YAP-miRFP670 and mScarlet-I-SOX2, tracked from the first cell division through the 32-cell stage. Expression of mScarlet-I-SOX2 intensity (minmax normalized from 0 to 1) is color-mapped in purple onto the branches of the lineage tree. t=0 is defined as the average time of division from the 8- to 16-cell stage. Unit of time is hours. (F) Time-lapse images of a representative embryo expressing YAP-miRFP670 and CDX2-eGFP, tracked from the first cell division through the 32-cell stage. Minmax normalized CDX2-eGFP intensity is color-mapped in green.

acquired a view of the simultaneous dynamics of nuclear YAP and its downstream targets within lineages of all blastomeres. To model variability in the observed dynamics, we chose a dynamic Bayesian network approach, which is a well-established strategy for modeling gene expression time series (Kim et al., 2003; Robinson and Hartemink, 2008; Koller and Friedman, 2009; Ruiz-Perez et al., 2021; Suter et al., 2022). The resulting generative model reveals the dynamics and variability of the first cell fate choice in mammalian development.

## RESULTS

### Live imaging YAP and its target genes

Taking advantage of the simplicity of the network that controls the first mammalian cell fate choice, as well as the small cell number at which this choice takes place, we sought to characterize the dynamics of YAP and its targets by live imaging. As the first step towards this goal, we established or used already existing fluorescent reporters of YAP (YAP-miRFP670; Gu et al., 2022), CDX2 (CDX2-eGFP; McDole and Zheng, 2012) and SOX2 (mScarlet-I-SOX2; this study). To generate a SOX2 reporter mouse line, we targeted mScarlet-I to the N-terminus of the endogenous *Sox2* allele (Fig. S1A). mScarlet-I expression displayed the expected pattern of endogenous SOX2 based on previous analysis of fixed samples, with expression initiating in a few cells at the late 16-cell stage and mScarlet-I present in all ICM cells at the blastocyst stage (Fig. 1C, Fig. S1E,F). Furthermore, treatment of embryos with the Rho-associated kinase (ROCK) inhibitor caused an increase in the number of cells expressing mScarlet-I-SOX2, in agreement with a previous report (Frum et al., 2018; Fig. S1B-D). These data indicate that the established reporter is suitable for monitoring endogenous SOX2 dynamics.

We used light sheet microscopy to image the joint dynamics of YAP-miRFP670 and CDX2-eGFP or YAP-miRFP670 and mScarlet-I-SOX2, along with H2B-miRFP720 to identify cell nuclei (Fig. 1C,D). Nuclei were then segmented and tracked using our previously established pipeline (Nunley et al., 2024) to construct complete lineage trees of developing embryos. To quantify the dynamics of YAP and its targets within each lineage, fluorescent intensities for each reporter were extracted from nuclear masks (Fig. 1E,F, see Materials and Methods). Every embryo in our dataset was imaged from the 8-cell stage to the end of the 32-cell stage, with the final cells defining 32 branches of expression histories. For each tree branch of every embryo, we sought to ultimately characterize the dynamics of YAP and its targets in the $(Y, S, C)$ phase space, where $Y$ is the nuclear concentration of YAP, and $S$ and $C$ are the total nuclear levels of SOX2 and CDX2, respectively.

Cells in mouse embryos exhibit random normally distributed timing of cleavages at the 16- and 32-cell stages (Fabrèges et al., 2024). We normalized the time for each branch of every embryo by warping it to align cell division times at every cell cycle (see Materials and Methods). The expression of both YAP targets was heterogeneous, both in the timing and levels of expression (Fig. 2A,B). This was accompanied by heterogeneity in the corresponding YAP trajectories.

### Coarse-graining dynamics

As the first step towards modeling the observed dynamics, we coarse-grained the phase space by distinguishing only two levels for each variable, which effectively views YAP as either cytoplasmic or nuclear, and each of its targets as either expressed or not expressed. To this end, we first separated 16- and 32-cell stages into early (E) and late (L) stages, resulting in five stages of interest: 8, 16E, 16L,

32E and 32L, indexed by $i=0…4$. For every stage, the expression levels for YAP, SOX2 and CDX2 were binarized (see Materials and Methods). In particular, YAP concentration was $z$-scored at every time point and averaged over each stage; by pooling the data, we defined universal stage-dependent thresholds (Fig. S2B) to characterize YAP as either nuclear (1) or cytoplasmic (0). To binarize the expression of each downstream TF, after appropriate normalization, we applied a stage-independent threshold to the values at the end of 16L, 32E and 32L to define positive (1) and negative (0) cells at these stages ($N=12$ embryos for SOX2, $N=9$ embryos for CDX2, Figs. S2C-E). We characterized YAP as nuclear for all cells at stage 8 and all cells as negative for both TFs at stages 8 and 16E.

As a result of coarse-graining, imaging data from every embryo was decomposed into 32 binarized trajectories indexed by stage: $i=0…4$. More precisely, YAP dynamics and expression of its downstream gene $g$ were each described by binary sequences of length 5, $Y=\{Y_i\}_{i=0…4}\in\{0, 1\}^5$ and $G=\{G_i\}_{i=0…4}\in\{0, 1\}^5$ for $G=S, C$ (Fig. 2A,B). Coarse-graining allowed us to classify the trajectories by their time of expression induction, i.e. $\min\{i : G_i = 1\}$. For $G=S, C$, we denoted 'induction classes', i.e. the classes of trajectories inducing expression at 16L, 32E and 32L stages, by $G_{16L}^+, G_{32E}^+$ and $G_{32L}^+$, respectively. Note that, after induction, a trajectory can potentially switch back to 0. Such loss of expression was very rare, though possible, for both TFs (see Fig. 2A for a SOX2 example). The rest of the trajectories never induced the TF, and we denoted the corresponding induction class by $G^-$. We observed that trajectory classes might exhibit different YAP coarse-grained dynamics (Fig. 2A,B). To systematically explore the role of YAP in setting the times at which cells express SOX2 and CDX2, we proceeded to model the coarse-grained dynamics in $(Y, G)$ phase space using dynamic Bayesian networks.

### Bayesian modeling of pairwise dynamics

Bayesian networks model stochastic systems using directed graphs with random variables in the vertices. The edges encode the relationships between the nodes, with each node associated with a conditional probability distribution given its parents. This structure allows to factorize the joint probability distribution of the nodes and simplify inference (see Materials and Methods; Koller and Friedman, 2009). In our case, each node of the graph corresponds to the level of nuclear YAP, $Y_i$, or one of its targets, $G_i$, at a time point $i$. To model temporal dependencies, we restricted the parents of each node to those at the same or previous time point. Thus, we adopted a dynamic Bayesian network framework, with $\{Y_i\}_{i=0…4}$ and $\{G_i\}_{i=0…4}$ in the nodes.

After binarizing all variables, we modeled each node using a Bernoulli distribution. YAP localization was modeled with a linear chain structure. Assuming no prior information on the timescales of molecular processes linking subcellular YAP localization to the expression of its targets, we explored several candidate model architectures downstream of YAP (Fig. 3A). The simplest network assumed a fast effect of YAP on downstream protein concentrations, captured by $\{(Y_i, G_i)\}_{i=0…4}$ edges, with no additional dependencies between the $G$ nodes (M1 model). We also considered networks in which $G$ nodes were connected by another chain, representing additional dependencies in the target expression. These models include fast (M2) or slow, captured by $\{(Y_i, G_{i+1})\}_{i=0…3}$ edges, timescales for the effect of $Y$ on $G$ (M3). Finally, we included a network that combined the fast and slow timescales (M4). All models were treated as non-homogeneous, i.e. the conditional probability distributions were allowed to vary across time points. To

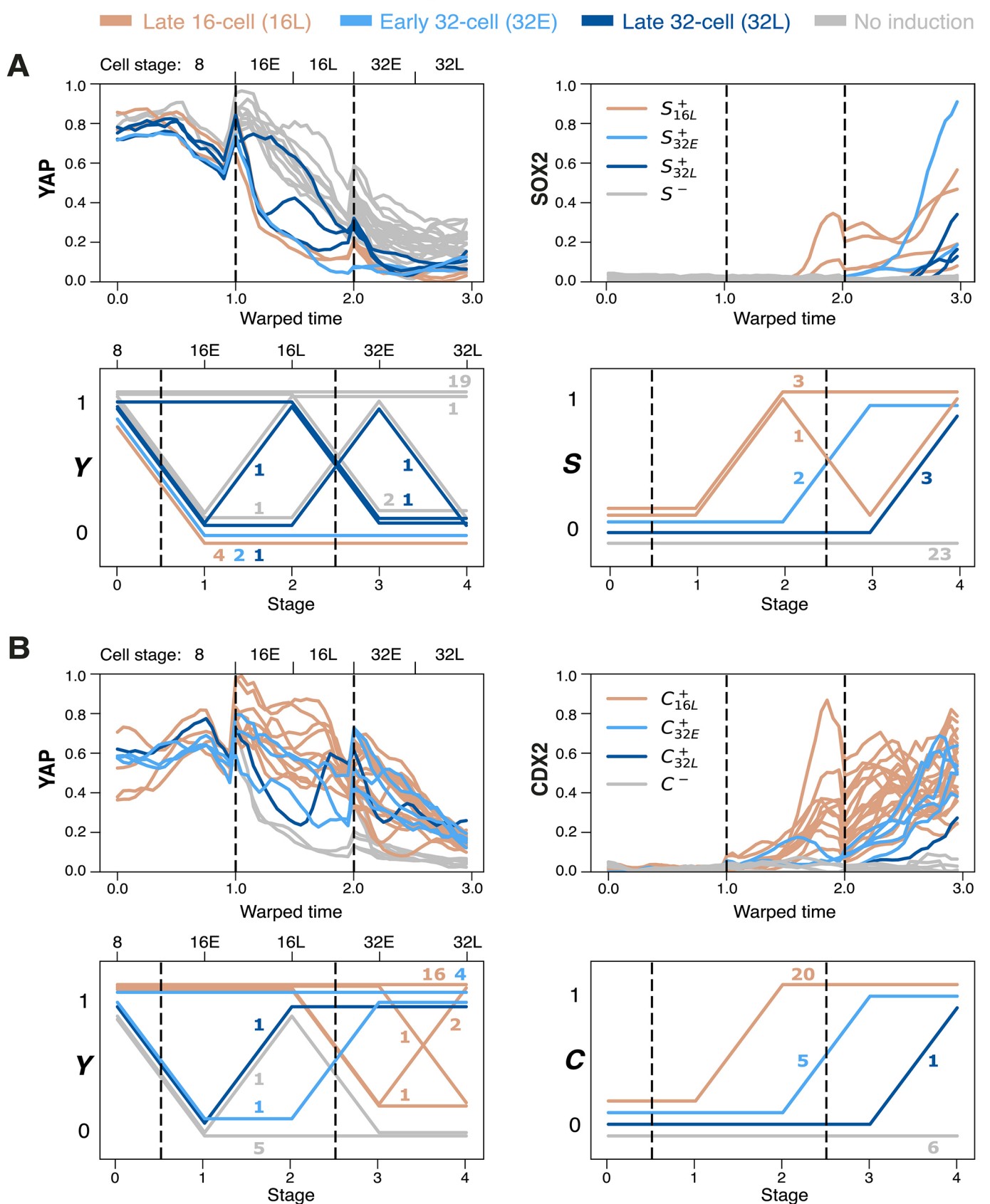

**Fig. 2.** See next page for legend.

**Fig. 2. Dynamics of nuclear YAP concentration and expression of SOX2 and CDX2.** (A) Extracted and discretized dynamics of YAP localization and SOX2 expression in the cell lineages of one representative embryo. Imaging starts at the 8-cell stage and all cells undergo two division rounds. Each branch is colored according to its induction class ($S_{16L}^+$, $S_{32E}^+$, $S_{32L}^+$ and $S^-$), i.e. the stage of SOX2 induction. See below for the description of stages. Top left and right, x-axis: linearly warped time, with division times aligned for all branches. Top left, y-axis: average YAP reporter intensity in segmented nuclei, smoothed and minmax normalized between 0 and 1 (see Materials and Methods). Top right, y-axis: total SOX2 reporter intensity in each segmented nucleus, smoothed and minmax normalized between 0 and 1 (see Materials and Methods). Bottom left and right, x-axis: 5 stages [0(8), 8-cell stage; 1(16E), first half of 16-cell stage; 2(16L), second half of 16-cell stage; 3(32E), first half of 32-cell stage; 4(32L), second half of 32-cell stage]. Black dashed vertical lines indicate 8/16 cell and 16/32 cell division times. Bottom left, y-axis: average YAP reporter intensity in segmented nuclei, smoothed and discretized for every stage (see Materials and Methods). Bottom right, y-axis: total SOX2 reporter intensity in segmented nuclei, smoothed and discretized for every stage (see Materials and Methods). Thirty-two branches of YAP localization histories are shown, grouped by their YAP behavior and induction class. Number of branches in each group is enumerated next to the discretized trajectory. (B) As A for YAP and CDX2 in a different representative embryo. Each branch is colored according to its stage of CDX2 induction.

summarize, we chose four candidate models (M1-M4) to be evaluated on every dataset.

For each $(Y, G)$ pair, we sought to select the best model out of the four candidates. With every embryo decomposed into 32 trajectories as described above, we could fit every candidate model to hundreds of observations (trajectories) in the $(Y, G)$ phase space. Specifically, the YAP-CDX2 dataset included $N=9$ embryos ($n=288$ observations) and the YAP–SOX2 dataset included $N=12$ embryos ($n=384$ observations).

We performed model selection using the Bayesian Information Criterion (BIC). Each dataset was randomly split into independent training and testing sets, with each training set consisting of six embryos. BIC leverages maximum likelihood estimation while penalizing models for the number of parameters, reducing the risk of overfitting. Interestingly, the same network architecture (M3, Fig. 3A) was selected on the training set via BIC for both TFs (Fig. S3A). Moreover, M3 remained the top model across a broad range of TF discretization thresholds (Fig. S3C), which led us to select this model for subsequent analyses. This network suggests a delayed regulatory effect of YAP localization on nuclear TF levels at the timescale of several hours (half cell cycle length, Fig. S2A). M3 also includes dependence of TF concentration on its previous state, capturing temporal persistence in target expression. Thus, the winning network structure allows to formulate data-driven hypotheses on the timescales of regulation of SOX2 and CDX2 by YAP.

With the ultimate aim to apply M3 to predict the distribution of trajectories over induction classes ($G_{16L}^+$, $G_{32E}^+$, $G_{32L}^+$, $G^-$) for each TF, we first tested the model for robustness, observing stable predictions between different train-test splits (see Materials and Methods, Fig. S3D). Furthermore, on average across multiple train-test splits, the models demonstrated good generalization performance on held-out test data (see Materials and Methods, Fig. S3E). Therefore, we could proceed with applying the M3 model to the full datasets.

### Transition dynamics of the top-performing model

Observations for the pairwise datasets could be summarized at every stage (Fig. 3B). The dynamic nature of the data, however, allowed us to extract dependencies between the variables at adjacent stages and fit the transition matrices that fully define each model. There are

1024 $(Y, G)$ trajectories that are theoretically possible, and the joint probability distribution provided by the network assigns a probability to each of them. More precisely, the probability distribution over trajectories is decomposed into the product of the initial probability distributions $p(Y_0)$ and $p(G_0)$, and transition probabilities of the form $p(Y_{i+1}|Y_i)$ and $p(G_{i+1}|G_i, Y_i)$ for $i=0…3$. The initial condition for $Y$ is always fixed at 1, giving $p(Y_0=1)=1$ with $p(Y_{i+1}|Y_i)$ describing YAP localization dynamics for later stages, e.g. the transition matrix:

$$p(Y_1|Y_0) = \begin{array}{c|cc} & \multicolumn{2}{c}{Y_1} \\ Y_0 & 0 & 1 \\ \hline 1 & 0.32 & 0.68 \end{array} \qquad (1)$$

contains in its columns the probabilities of excluding YAP from the nucleus $[p(Y_1 = 0|Y_0 = 1) = 0.32]$ and retaining it $[p(Y_1 = 1|Y_0 = 1) = 0.68]$ at the 8- to 16-cell state division. At the 8/16 division, it is known that most cells divide asymmetrically, giving rise to one cell with nuclear YAP and one cell with cytoplasmic YAP. The transition matrix in Eqn 1 corresponds to a 64% rate of asymmetric division, which is in agreement with available data (Yamanaka et al., 2010; Anani et al., 2014; Watanabe et al., 2014; McDole et al., 2011). We obtained analogous probabilities of YAP state transitions between every two consecutive time points.

At the same time, both TFs are not initially expressed, fixing their initial condition at 0, $p(G_0=0)=1$. We fit the transition matrices for every $G_i$ node and its parents. A simple example is provided by the SOX2 transitions at the early-to-late 32-cell stage:

$$p(S_4|S_3, Y_3) = \begin{array}{cc|cc} & & \multicolumn{2}{c}{S_4} \\ S_3 & Y_3 & 0 & 1 \\ \hline 0 & 0 & 0.57 & 0.43 \\ 0 & 1 & 0.97 & 0.03 \\ 1 & 0 & 0 & 1 \\ 1 & 1 & 0 & 1 \end{array} \qquad (2)$$

From this matrix, one can conclude that, for cells with nuclear YAP ($Y_3=1$) and for SOX2$^+$ cells at the 32E stage ($S_3=1$), the state of SOX2 propagates to the 32L stage with probability close to 1. However, for SOX2− cells with cytoplasmic YAP at the 32E stage ($S_3=0$, $Y_3=0$), SOX2 becomes induced by the 32L stage ($S_4=1$) in a non-deterministic fashion, with probability 0.43. Non-deterministic elements (transition probabilities away from 0 or 1) were also contained in transition matrices at other stages for the YAP-SOX2 model, as well as for some transition matrices of the YAP-CDX2 model. This feature observed in both models provides another motivation for our probabilistic modeling approach.

Although some progress in probing YAP-CDX2 and YAP-SOX2 associations is possible by direct analysis of the observed trajectories, modeling can be used for inference. Bayesian modeling can be used to calculate posterior probabilities exactly or approximately, via sampling. As we illustrate below, our trained Bayesian models can be used to augment the empirical dataset by synthetic trajectories consistent with the inferred transition matrices. Such use of Bayesian models for statistical analysis of system dynamics is common in other contexts (van de Schoot et al., 2021; Griffiths et al., 2024) and could be applied here to samples from the posterior distribution of any embryonic statistic.

### Model-based inference

We used Bayesian networks trained on pairwise dynamic data for $(Y, G)$ to generate synthetic embryos. Every embryo in our synthetic dataset starts with 8 cells, each undergoing two rounds of division

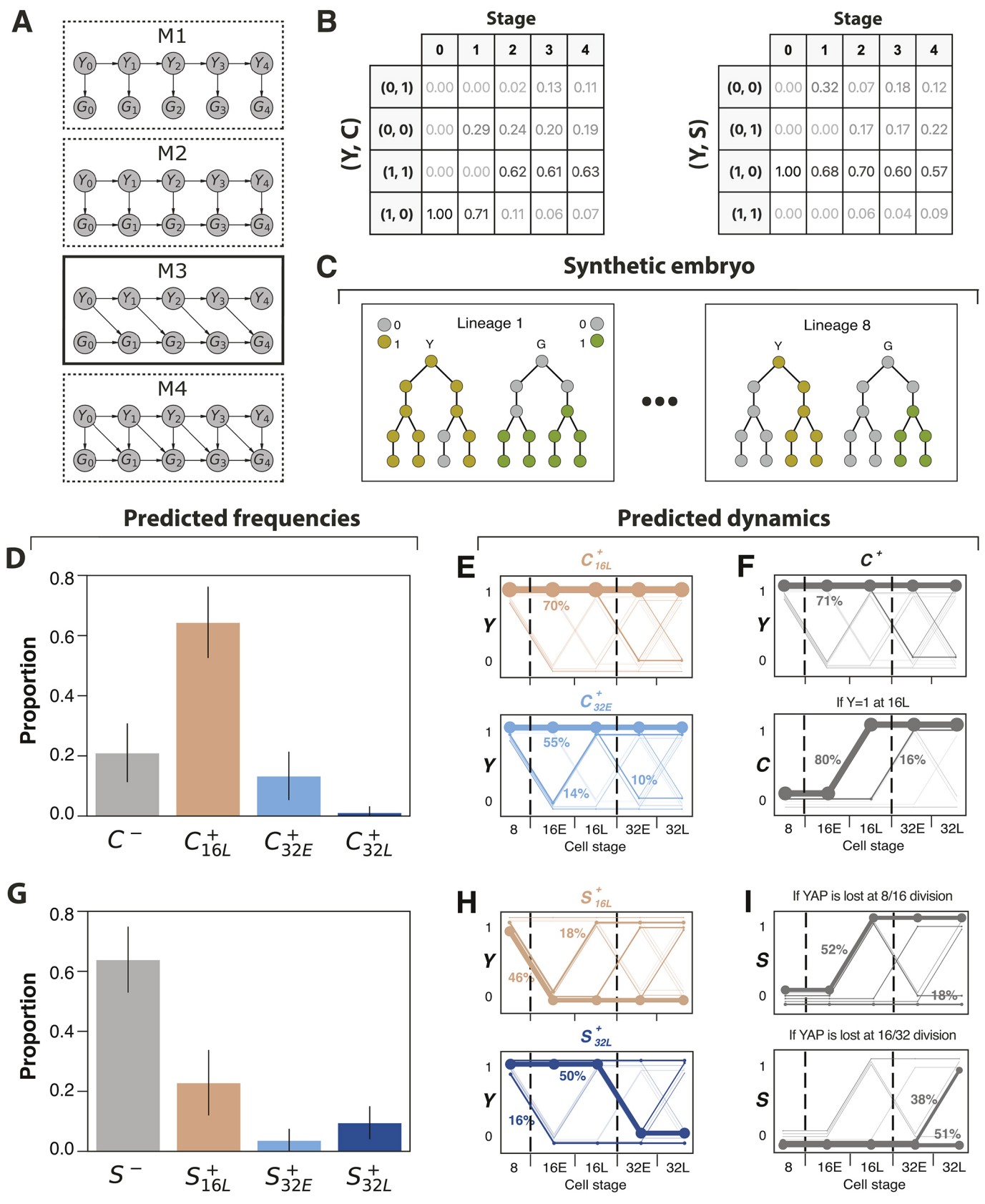

**Fig. 3.** See next page for legend.

over 5 stages resulting in 32 cells at the final stage (Fig. 3C, see Materials and Methods). An embryo comprises 8 independently sampled lineage trees, each starting with one cell sampling from the initial distribution $p(Y_0, G_0)=p(Y_0)p(G_0)$. The cells in each tree undergo transitions in the $(Y, G)$ phase space, sampling from the corresponding transition matrices. At divisions, expression values

**Fig. 3. Modeling gradual induction of CDX2 and SOX2 with dynamic Bayesian networks.** (A) Bayesian network architectures considered. $Y_i$: binarized nuclear YAP concentration at stage $i=0…4$. $G_i$: total nuclear level of a TF at stage $i$. $G=C$ for CDX2 and $G=S$ for SOX2. The winning M3 model is shown with the solid outline. (B) Table summarizing frequencies of binarized ($Y$, $G$) observations at every stage. Left: CDX2, $N=9$ embryos. Right: SOX2, $N=12$ embryos. Eight observations (cells) per embryo for $i=0$; 16 cells per embryo for $i=1$, 2; 32 cells per embryo for $i=3$, 4. (C) A schematic of an embryo simulated from a Bayesian network (see Materials and Methods). A synthetic embryo is a collection of 8 independent lineage trees (sample trees for lineages 1 and 8 are shown). Every lineage comprises 13 synthetic cells: one cell undergoing 2 rounds of division over 5 stages. For a lineage, left: nuclear YAP ($Y=1$) is shown in olive, cytoplasmic YAP ($Y=0$) in gray; right: gene expressed ($G=1$) is shown in green; gene not expressed ($G=0$) is shown in gray. (D) Frequencies of CDX2 induction classes over 3000 simulated embryos. Bar, mean frequency; error bar, one standard deviation around the mean. (E) Top: posterior distribution of $Y$ for cells inducing CDX2 at 16L stage (i.e. conditioned on $C_2=1$). Linewidth of a trajectory is proportional to its posterior probability. Posterior probabilities are marked next to the corresponding most-prominent branches. Bottom: posterior distribution of $Y$ for cells inducing CDX2 at 32E stage. (F) Top: posterior distribution of $Y$ for cells inducing CDX2 for all CDX2+ cells ($C_4=1$). Bottom: posterior distribution of $C$ trajectory for cells with nuclear YAP at 16L stage (i.e. conditioned on $Y_2=1$). (G) Frequencies of SOX2 induction classes over 3000 simulated embryos. (H) Posterior distribution of $Y$ for cells inducing SOX2 at 16L stage (top) or inducing SOX2 at 32L stage (bottom). (I) Top: posterior distribution of $S$ for cells losing nuclear YAP at the 8/16 cell division (i.e. conditioned on $Y_1=0$). Linewidth of a trajectory is proportional to its posterior probability. Posterior probabilities are marked next to the corresponding most-prominent branches. Bottom: posterior distribution of $S$ for cells losing nuclear YAP at the 16/32 cell division ($Y_2=1$, $Y_3=0$).

for daughter cells are independently sampled conditional on their mother. Every synthetic embryo could thus be viewed as an ensemble of 32 trajectories of expression histories of cells at the 32L stage and their predecessors. These trajectories are pairwise independent for cells from different lineage trees but are not necessarily independent within one lineage. Ultimately, this approach can be applied to any dynamic Bayesian network to simulate expression dynamics of any number of variables, taking into account lineage relationships.

Since YAP is a key regulator for both CDX2 and SOX2, it is natural to ask whether particular features of YAP dynamics are a necessary or sufficient condition for induction of these TFs. Furthermore, as shown above, the expression dynamics of YAP targets in different cells can be classified by the timing of their induction. We asked whether the cells that belong to different classes could be distinguished by differences in their YAP localization time courses. If YAP were the sole activator of CDX2, it could be expected that early expression of CDX2 would be associated with more persistent levels of nuclear YAP. In contrast, one might expect that the first cells to express SOX2 should be characterized by early loss of nuclear YAP. Note that similar questions naturally arise whenever one regulator controls several targets, e.g. in the context of tissue patterning by morphogen gradients (Ashe and Briscoe, 2006).

We used probabilistic inference to investigate variability in the induction timing of YAP targets, and the extent to which this variability can be explained by differences in nuclear YAP. We first generated thousands of 32L embryos describing dynamics in the ($Y$, $G$) phase space, independently for $G=S$ and $G=C$. Each trajectory in an embryo belongs to one of the $G_{16L}^+$, $G_{32E}^+$, $G_{32L}^+$ and $G^-$ classes indicating its time of induction (see 'Live imaging YAP and its targets' for a definition). Thus, we could estimate the expected frequency of each class for $G$, as well as the inter-embryonic variability of the frequencies.

Analyzing the simulated data, we found that CDX2 is most frequently induced at 16L or 32E stages (Fig. 3D), at proportions 0.64±0.12 and 0.13±0.08, respectively. Trained Bayesian networks were then used to predict the joint posterior distribution of $Y$ dynamics of a cell/trajectory, conditional on its induction class (Fig. 3E, see Materials and Methods). Interestingly, we observed little difference in the time courses of nuclear YAP in cells that induced CDX2 at the 16L and 32E stages, with both distributions mainly concentrating on cells with consistently nuclear YAP (70% and 55%, respectively). Therefore, YAP dynamics cannot be reliably used as a predictor of CDX2 induction timing (see Fig. S3F for a representative time course). In CDX2+ cells ($C_4=1$), 71% were predicted to have consistently nuclear YAP, with 96% of the trajectories having nuclear YAP at 16L stage (Fig. 3F), identifying nuclear YAP at this stage as a necessary condition for CDX2 expression by 32L stage. At the same time, nuclear YAP at 16L stage was predicted to result in CDX2 induction with overwhelming probability, making it a sufficient condition for CDX2 expression by 32L stage.

For SOX2, induction was predicted to be most likely at 16L or 32L stages (0.23±0.11 and 0.1±0.05 frequency, respectively, Fig. 3G). Conditioning on SOX2 classes of induction, we found that, for most cells, early loss of nuclear YAP was a necessary condition for the early onset of SOX2 expression (Fig. 3H). More precisely, 93% $S_{16L}^+$ cells were predicted to have removed YAP from the nucleus at the 8/16 cell division. At the same time, most of $S_{32L}^+$ cells were also predicted to lose YAP at one of the divisions, with probability 0.23 at the 8/16 division and probability 0.63 at the 16/32 cell division. Thus, loss of nuclear YAP at a division is typically a necessary condition for SOX2 induction. However, trajectories losing nuclear YAP at each of the divisions can fail to induce SOX2 at the next stage (Fig. 3I). Specifically, after loss of nuclear YAP at the 8/16 division, cells do not induce SOX2 at 16L with probability 0.34, and after YAP loss at the 16/32 division, there is no SOX2 expression with probability 0.53. In summary, loss of YAP at a division is typically required but not sufficient for induction of SOX2. Most cells induce SOX2 at the second half of the subsequent cell cycle; however, some cells may delay SOX2 induction or never induce it by the 32L stage (see Fig. S3F for representative examples).

**Model-based data fusion**

Having trained Bayesian networks for pairwise dynamics, we could establish a framework for modeling dynamics in the joint ($Y$, $C$, $S$) phase space. Indeed, the M3 architectures of the ($Y$, $C$) and ($Y$, $S$) models could be fused via their common chain in $Y$ variable (Fig. 4A). With separate datasets for ($Y$, $C$) and ($Y$, $S$), we can train different parts of the network independently, accounting for missing data (see Materials and Methods). The fused network was used to generate synthetic ($Y$, $C$, $S$) lineages and simulate embryos comprising 8 lineage trees analogously to pairwise models. Expression of every simulated lineage could be viewed as a branching trajectory in the ($Y$, $C$, $S$) phase space, i.e. on a $\{0, 1\}^3$ cube (Fig. S4A). At every stage $i$, both $C_i$ and $S_i$ are binary, which led us to classify trajectories for every $i$ into 4 classes (states): $C^+ S^-$ if $C_i=1$, $S_i=0$; $C^- S^+$ if $C_i=0$, $S_i=1$; double positive, $C^+ S^+$, if $C_i=1$, $S_i=1$; and double negative, $C^- S^-$, if $C_i=0$, $S_i=0$.

We characterized the composition of a synthetic embryo at every stage $i$ by the numbers of cells in the four classes (Fig. S4B). Synthesizing embryos with the fused model, we could extract the joint probability distribution of cell state counts at every stage. In particular, we could sample from the marginal distributions of state counts at every stage (Fig. 4B). For example, at the final stage, the fused model predicts that one can expect 19.7±3.4 $C^+ S^-$ cells,

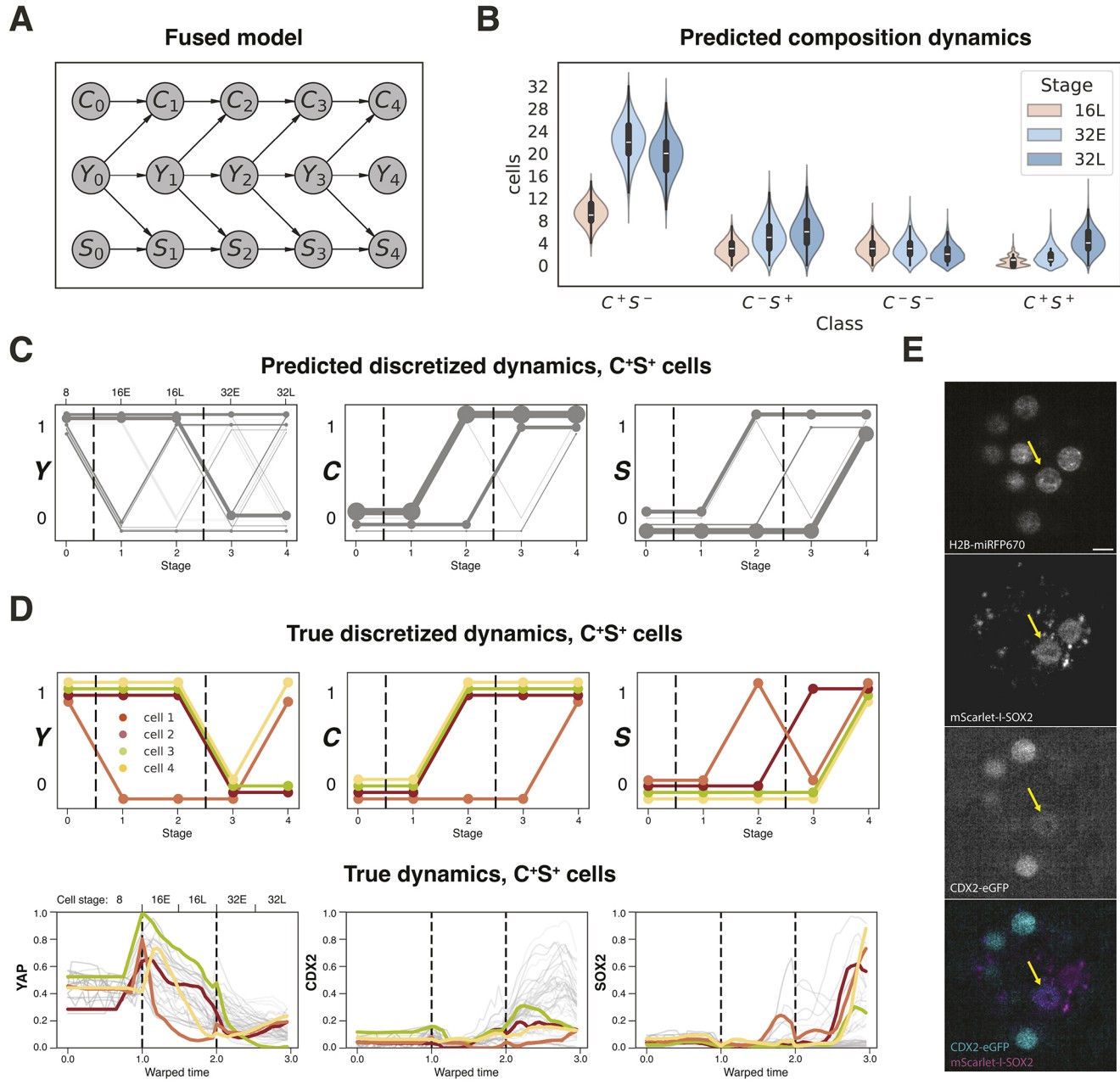

**Fig. 4. Modeling joint dynamics of YAP, CDX2 and SOX2 with a fused Bayesian network.** (A) The structure of the network that was used for data fusion: $C_i$, (binarized) nuclear CDX2 level at stage $i$; $S_i$, (binarized) nuclear SOX2 level at stage $i$; $Y_i$, (binarized) nuclear YAP concentration at stage $i$. (B) Violin plot showing posterior marginal distributions for the numbers of cells in four possible classes at 16L, 32E and 32L stages. Distributions are smoothed with kernel density estimation. 3000 embryos were simulated. (C) Left: posterior distribution of $Y$ for $C^+ S^+$ cells (i.e. conditioned on $C_4$=1, $S_4$=1). Linewidth of a trajectory is proportional to its posterior probability. Middle: posterior distribution of $C$ for $C^+ S^+$ cells. Right: posterior distribution of $S$ for $C^+ S^+$ cells. (D) Top: discretized trajectories of $Y$, $C$ and $S$ for 4 $C^+ S^+$ cells identified in 2 embryos simultaneously expressing YAP-miRFP670, CDX2-eGFP and mScarlet-I-SOX2. Bottom: nuclear YAP concentration (left), total nuclear levels of CDX2 (middle) and total nuclear levels of SOX2 (right) for the same 4 cells as above. Data from all other cells in the 2 embryos are shown in gray. Values on $y$-axis were smoothed and minmax normalized between 0 and 1 within every embryo. $x$-axis: linearly warped time, with division times aligned. Black dashed vertical lines: 8/16 cell and 16/32 cell division times. (E) Light-sheet microscopy images of an embryo simultaneously expressing YAP-miRFP670, CDX2-eGFP, mScarlet-I-SOX2 and H2B-miRFP720. Arrows indicate a cell that was classified as $C^+ S^+$. A $z$-slice is shown. Scale bar: 10 µm.

$6\pm2.8 C^- S^+$ cells, $2.2\pm1.5 C^- S^-$ cells and $4.2\pm2.0$ $C^+ S^+$ cells in an embryo.

Interestingly, the fused model predicted the appearance of double-positive $C^+ S^+$ cells by the 32L stage in most embryos. Furthermore, we could extract the posterior marginal distributions of $Y$, $C$ and $S$ in the $C^+ S^+$ class (Fig. 4C). While different YAP trajectories can give rise to a $C^+ S^+$ cell at the 32L stage, this

behavior was predicted to most frequently result from loss of YAP at the 16/32 division (with $Y$=[1, 1, 1, 0, 0] with 32% probability). Sixty-two percent of the $C^+ S^+$ cells were predicted to induce CDX2 at $i$=2, the 16L stage. At the same time, we saw SOX2 induction at $i$=4, the 32L stage, with a probability of 53%. Moreover, fusing the network with an analogous Bayesian network modeling the relationship of nuclear YAP concentration with cell position

(Chalifoux et al., 2025; 'geometric model') allowed us to predict position dynamics of double-positive cells (Fig. S4C). In the geometric model, cell position was captured via its relative exposed surface area, defined as the proportion of the surface area of the cell that is not in contact with other cells of the embryo. With three possible positions defined at every stage, inner, intermediate and outer, we found that $C^+ S^+$ cells are most likely to assume the inner position by the 32L stage.

To validate the presence of double-positive cells, we imaged the joint dynamics of YAP-miRFP670, CDX2-eGFP and mScarlet-I-SOX2, along with H2B-miRFP720, in two embryos expressing all four reporters. Using the same pipeline for segmentation, signal extraction and discretization of each regulator as before, we identified 4 $C^+ S^+$ cells in these embryos (see Fig. 4E for a representative example). Furthermore, tracking one of these embryos to the end of the 64-cell stage allowed us to observe resolution of double-positive expression in the progeny of a $C^+ S^+$ cell, with both daughter cells maintaining SOX2 expression and losing CDX2 expression (Fig. S4D). We then compared the discretized profiles of $Y$, $C$ and $S$ in these embryos (Fig. 4D) to our predictions. While the identified $C^+ S^+$ cells indeed demonstrated variable $Y$, $C$ and $S$ trajectories, more frequent trajectories for each variable coincided with the modes of the corresponding posterior marginal distributions predicted by the model in $C^+ S^+$ cells, further validating our approach.

## DISCUSSION

The molecular components and cellular processes involved in the first cell fate bifurcation in mammalian development have been mainly elucidated based on data from fixed embryos. However, understanding the expression dynamics of this bifurcation requires access to lineages of individual blastomeres, which is impossible to obtain from fixed embryos, due to the well-recognized variations in blastomere arrangements and other sources of gene expression variability at this stage. Here, we present a live imaging and cell tracking approach the reveals the dynamics of nuclear YAP and its transcriptional effects across cell cycles. We also demonstrate that the time series extracted from blastomere lineages enable probabilistic modeling and model-based inferences about the dynamic process of cell allocation to the ICM and TE compartments.

Our modeling approach relies on the formalism of dynamic Bayesian networks. Bayesian networks provide a decomposition of the joint probability distribution of the nodes, which can be used to calculate posterior probabilities and test specific hypotheses. Using this approach, we analyzed the necessity and sufficiency of specific features of YAP dynamics for the expression of two downstream targets: CDX2 and SOX2. Our model predicted that nuclear YAP at 16L stage is both required and sufficient for CDX2 expression; however, YAP was not a good predictor of the exact timing of expression onset. This indicates that additional modifiers are likely variable in cells that contribute to the regulation of CDX2 expression alongside YAP. A possible candidate is the Notch pathway, which has been shown to regulate *Cdx2* in parallel to Hippo signaling (Watanabe et al., 2017; Rayon et al., 2014; Menchero et al., 2019).

In the case of SOX2 we find that removal of YAP from the nucleus is typically required for SOX2 expression. As YAP removal mainly occurs at either the 8/16 or the 16/32 cell divisions and SOX2 expression follows with some delay, we find that SOX2 is mostly activated in two waves: at the 16L and 32L cell stages. Interestingly, however, YAP removal is not sufficient for SOX2 expression, as we find significant stochasticity in whether SOX2 is induced or not in these cells. Indeed, SOX2 was shown to be

prematurely activated in only a subset of cells of 8-cell stage embryos lacking TEAD4 or YAP (Frum et al., 2019). These observations highlight that we are lacking critical regulators of SOX2 in the embryo, either an additional repressor or an activator, the activity of which is highly heterogeneous among ICM cells. Notably, factors such as OTX2 and CARM1 have been suggested to regulate Sox2 expression in the embryo (Acampora et al., 2016; Torres-Padilla et al., 2007). Whether endogenous heterogeneities in these factors can explain SOX2 variabilities will need to be investigated in the future. The timing of ICM fate establishment has been shown to have implications for the subsequent cell fate decision, when ICM cells differentiate into epiblast and primitive endoderm cells, with early specifying ICM cells displaying a bias towards the epiblast fate (Mihajlović et al., 2015; Morris et al., 2013, 2010; Krupa et al., 2014; Yamanaka et al., 2010). SOX2, in particular, may have important roles in this process, as it is an upstream regulator of *Fgf4* expression, a key ligand produced by epiblast cells (Mistri et al., 2018). Therefore, revealing the sources of variability in the timing of ICM fate acquisition and, specifically, understanding the drivers of SOX2 induction are important for investigating the dynamics of the next cell fate decision.

Bayesian networks can account for missing data and can be used for data fusion (Koller and Friedman, 2009). We have demonstrated this by integrating pairwise observations into the joint description of YAP, CDX2 and SOX2. This allowed us to reveal the frequencies and origins of double-positive cells in the embryo. These cells most frequently arise from early CDX2 induction, followed by loss of nuclear YAP (likely an internalization event) and subsequent induction of SOX2. This is in agreement with previous observations of CDX2-positive inner cells resulting from internalization events (Toyooka et al., 2016; McDole and Zheng, 2012), which will likely completely downregulate CDX2 and assume an ICM identity. In the future, this data fusion approach can be extended to synthesize trajectories that include other important variables, such as cell shape and polarity, and cell cycle progression.

## MATERIALS AND METHODS
### Transgenic mouse lines
*Yap-miRFP670* (Gu et al., 2022), *mScarlet-I-Sox2* (generated in this study) and *Cdx2-eGFP* (McDole et al., 2011) mouse lines were used in this study. All mice were bred on a CD-1 (ICR) (Charles River) background. All animal work was carried out at Princeton University, according to the National Research Council's Guide for the Care and Use of Laboratory Animals. Animal maintenance and husbandry were carried out in accordance with the Laboratory Animal Welfare Act. All procedures were approved by Princeton University's Institutional Animal Use and Care Committee (IACUC protocol 2133). Mice were housed in a 21°C facility with a 14 h daily light cycle and 48% average ambient humidity.

### Generation and validation of *mScarlet-I-Sox2* reporter mouse line
The *mScarlet-I-Sox2* targeting vector was custom synthesized (Qinglan Biotech). The targeting vector consists of a 988 bp 5′ homology arm (the start codon and the genomic sequence immediately upstream of the start codon of *Sox2*), the coding sequence of mScarlet-I, a linker sequence (3×GGGGS) and an 800 bp 3′ homology arm (the genomic sequence immediately downstream of the start codon of *Sox2*) in a pMV backbone. The targeting plasmid was prepared using an endotoxin-free Maxi prep kit (Macherey-Nagel, NucleoBond Xtra Maxi EF, 740424.50). A sgRNA targeting the sequence around the start codon of *Sox2* was designed using CRISPOR (https://crispor.gi.ucsc.edu/) and synthesized (Synthego): *Sox2* N-term sgRNA, CGCCCGCATGTATAACATGATGG (TGG is the PAM). For details of the targeting strategy, see Fig. S1A.

Transgenic mice were generated by 2-cell cytoplasmic microinjection using the 2C-HR-CRISPR method (Gu, 2018). To obtain embryos for

microinjection, 5- to 6-week-old female CD-1 (ICR) (Charles River) mice were superovulated via intraperitoneal injection with 5 IU pregnant mare serum gonadotropin (PMSG, Sigma, G4527), followed by 5 IU human chorionic gonadotropin (hCG, Sigma, 9002-61-3) 47 h later. Superovulated females were then mated with 8- to 20-week-old CD-1 males and embryos derived from this cross were isolated at E1.5 (2-cell) by dissecting and flushing oviducts. Microinjection of E1.5 embryos was performed in M2 media (CytoSpring, M2115) in an open glass chamber using a Leica DMi8 inverted microscope equipped with a FemtoJet (Eppendorf), micromanipulators (Leica Microsystems) and a pinpoint electroporator device (micro-ePore, World Precision Instruments). Microinjection mixes were prepared in nuclease-free injection buffer (10 mM Tris-HCl at pH 7.4 and 0.25 mM EDTA) with 30 ng/μl targeting plasmid (as described above), 100 ng/μl *Cas9* mRNA and 50 ng/μl sgRNA. (See below for details on *Cas9* mRNA and sgRNA microinjection reagents.) Microinjected embryos were immediately transferred to pseudopregnant females via surgical oviduct transfer and gestated until birth. An ear biopsy of the pups was obtained at 2 weeks of age, genomic DNA was isolated using and Extract-N-Amp kit (Sigma, XNAT2-1KT) and genotyping PCR was used to identify founder mice containing the *mScarlet-I-Sox2* edit. Primers used were TCCCACAACGAGGACTACAC and CTTCAGCTCCGTCTCCATCA at an annealing temperature of 60°C. Founders were then crossed to CD-1 mice to obtain the N1 generation mice, which were genotyped using PCR and Sanger sequenced for correct integration of the reporter construct. N1 mice identified as heterozygous were bred together; however, only heterozygous offspring were obtained, indicating that this transgenic line could only be maintained in the heterozygous state. *mScarlet-I-Sox2* heterozygous animals were fertile and did not present any obvious phenotype. For validation of mScarlet-I-SOX2 expression, embryos were treated for 48 h, from E1.5 (2-cell) through E3.5 (blastocyst) with 20 μM ROCKi (Sigma, Y0503) in KSOM Embryomax (Sigma, MR-101-D) under LifeGlobal paraffin oil (CopperSurgical, LGPO-500) at 37°C, 5% $O_2$ and 5% $CO_2$.

### Reagents for microinjection

*In vitro* transcription of mRNA was performed as described previously (Gu, 2018). Briefly, pCS2-H2B-miRFP720 plasmid (cloned into pCS2 vector using miRFP720 sequence obtained from Addgene plasmid #136560) or pCS2-Cas9 plasmid (Addgene plasmid #122948) was linearized with NotI (New England Biolabs, R3189L) digestion and mRNA was synthesized using an mMessage mMachine SP6 intro transcription kit (Thermo Fisher Scientific, AM1340). mRNA was purified using an RNeasy Cleanup kit (Qiagen, 74104). mRNA was eluted into RNAse-free water and stored at −80°C.

### Embryo isolation and microinjection for imaging

For live imaging experiments, 2-cell stage (E1.5) embryos were isolated from *YAP-miRFP670*; *mScarlet-I-Sox2* females crossed with *YAP-miRFP670* males, from *YAP-miRFP670* females crossed with *YAP-miRFP670*; *Cdx2-eGFP* males, or from *YAP-miRFP670*; *mScarlet-I-Sox2* females crossed with *YAP-miRFP670*; *Cdx2-eGFP* males after natural mating. Females were 6-15 weeks old and males were 8-20 weeks old. Oviducts were flushed with M2 media (Zenith Biotech, M2116) and embryos were washed through microdrops of M2 under LifeGlobal paraffin oil (LGPO) (CopperSurgical, LGPO-500) before transferring to pre-calibrated microdrops of KSOM EmbryoMax (Sigma, MR-101-D) under LGPO in a 37°C, 5% $CO_2$ incubator for culture. Both cells of 2-cell stage embryos were microinjected in M2 media in an open glass chamber with 75 ng/μl H2B-miRFP720 mRNA. The same microinjection setup was used as described above for CRISPR genome editing of embryos. Injection mixes were prepared in a nuclease-free injection buffer (10 mM Tris-HCl at pH 7.4 and 0.25 mM EDTA). After injection, embryos were cultured in KSOM under LGPO until the start of imaging.

### Light-sheet microscopy

Light-sheet time lapse images were acquired on an inverted light sheet microscope (InVi from Luxendo/Bruker). The microscope was outfitted with an incubation chamber maintained at a constant temperature of 37°C, 5% $CO_2$, 5% $O_2$ and 95% relative humidity throughout the duration of the imaging. Individual embryos resided in ~100 μm deep wells made by gently pressing a blunt capillary tip into the base of a v-shaped chamber made of Fluorinated

Ethylene Propylene (FEP) foil [TruLive dish, 80-0031-02-00 (Luxendo/Bruker L-TLD-96)]. Approximately 6-15 embryos are loaded into the chamber for each experiment, each residing in its own ~100 μm deep well. Z-stacks for each embryo are acquired sequentially in up to four consecutive channels every 15 or 30 min for a total duration of ~30 h. Z-stacks of YAP-miRFP670 and H2B-miRFP720 were acquired every 15 min, while z-stacks of mScarlet-I-SOX2 and CDX2-eGFP were acquired every 30 min. Images were acquired with 2.0 μm z-axis resolution and 0.208 μm x- and y-axis resolution. Fluorescent signal was captured using the following laser wavelengths and filters: CDX2-eGFP, 488 nm excitation wavelength/497-554 BP emission filter; mScarlet-I-SOX2, 561 nm excitation wavelength/577-612 BP emission filter; YAP-miRFP670, 641 nm excitation wavelength/659-690 BP emission filter; H2B-miRFP720, 690 nm excitation laser/700LP emission filter.

### Image analysis of light-sheet microscopy data

Light-sheet microscopy data was analyzed following the image analysis pipeline outlined by Nunley et al. (2024). Briefly, light-sheet microscopy data was first converted using lossless compression to keller-lab-block (klb) format before being segmented for nuclei using a 3D-Stardist algorithm trained on preimplantation embryos (Nunley et al., 2024). The resulting nuclear regions of interest (ROIs) were checked and manually corrected, if necessary, in AnnotatorJ. Binary nuclear masks were then generated and used to register consecutive time frames using a modified MATLAB algorithm based on coherent point drift. Registered nuclear masks were then tracked through time in a semi-automated fashion. Average TF intensity was extracted from within the nuclear ROIs for each point in time. To correct for possible misalignment between the histone (hence nuclear masks) and TF channels, we performed rigid registration between the TF channel, aligning it to the histone channel. For that, we chose the transformation that minimizes the mean square deviation between their pixel intensities using matrix adaptation evolution strategy (Beyer and Sendhoff, 2017). After alignment, TF intensity extraction was performed over each nuclear mask using the MATLAB function 'regionprops3'. For every branch, the extracted signal was smoothed over a window of 2.5 h duration within every cell cycle.

### Image processing for figures and videos

For visualization purposes, maximum intensity projections (MIPs) or z-slices for all figures were processed in ImageJ by first cropping the field of view to contain the embryo, then adjusting brightness contrast settings to result in optimal visual contrast. Scale bars and timestamps were added in ImageJ. Images and videos were only modified for visualization purposes; all analysis was carried out on raw data.

### Time warping and coarse-graining

To normalize time for each individual branch for the length of the cleavage cycles, we linearly warped the time $t$ within every cycle between 0 and 1 ($\frac{t-\min_c t}{\max_c t-\min_c t}$ for cycle $c$) and rounded the times to the nearest multiple of 0.05 to form 20 time points per cycle. When more than one time point mapped to the same warped time, the average value was used. Warped times were linearly shifted according to the corresponding cleavage cycle, with [0,1], [1,2] and [2,3] corresponding to the 8-, 16- and 32-cell stages, respectively. To further coarse-grain time, we also used warped time to define early (E) and late (L) cycle stages by applying $x<0.5$ and $x>=0.5$ cutoffs on the warped times, respectively. This was applied to 16- and 32-cell stages only. As a result, five stages were defined: 8, 16E, 16L, 32E and 32L.

### Defining Y, S and C via discretization

We coarse-grained YAP, SOX2 and CDX2 dynamics, arriving at binary sequences $Y$, $S$ and $C$ indexed by stages, $i=0\ldots4$. The first 2 h post division were excluded from every branch, to normalize for small perturbations introduced by cell division. For YAP, we summarized dynamics for every branch using its average (after z-scoring) over each stage; for CDX2 and SOX2, we used their final values (after normalization) for every stage for every branch. For every stage, we then classified the branches into low (0) and high (1) expression based on these summaries. We restricted classification to 16E, 16L, 32E and 32L stages for YAP, and to 16L, 32E and 32L stages for the downstream TFs. In particular, at the 8-cell stage, i.e. for $i=0$, YAP is nuclear in all cells, and we assigned $Y_0=1$. At the same time,

both TFs are not initially expressed, and we put $S_0=S_1=0$ and $C_0=C_1=0$. For every variable $v$, we classified branches for every stage $i$ using thresholding on its stage-dependent summarized levels $\widetilde{v}_{b,i}$, with $b$ indexing the branches. The thresholds were chosen in a data-driven manner. In particular, for YAP, the thresholds were chosen to be stage dependent; for both TFs, the thresholds were chosen to be constant over stages. See below for details of discretization procedures and thresholds for each individual variable. Ultimately, for stage $i$ and variable $v$, a branch $b$ with expression above the corresponding threshold $\theta_i$, i.e. satisfying $\widetilde{v}_{b,i} > \theta_i$, was assigned $v_{b,i}=1$. Otherwise we put $v_{b,i}=0$.

### Discretization for nuclear YAP
In every embryo, we first z-scored YAP nuclear concentration at every time point, to normalize for the downward trends in its average nuclear concentration. Distributions of the summary variable $\widetilde{y}_{b,i}$, i.e. average z-scored concentration at stage $i$ where $b$ indexes the branches, were analyzed separately for every $i$. To separate the modes into low and high, we adopted the thresholding parameters derived by Chalifoux et al. (2025) for a dataset of $N$=13 embryos. We found that, at stages 16E and 16L, average YAP values exhibited bimodal distribution (Fig. S2B). In Chalifoux et al. (2025), we extracted a kernel density estimate for each distribution using the *distplot* method from the *seaborn* Python package, with default parameters, and identified its local minimum (via *scipy.signal.argrelmin*). For stages 32E and 32L, with no obvious bimodal structure present, we applied the same thresholding procedure as for CDX2 (see below). Thus, individual thresholds were applied to each half stage; $\theta_1=-0.59$ for 16E, $\theta_2=-0.82$ for 16L, $\theta_3=-0.61$ for 32E and $\theta_4=-0.58$ for 32L.

### Discretization for CDX2
For the analysis on CDX2 expression, total CDX2 signal in each nucleus was used. To classify individual branches using their CDX2 expression profiles (Fig. S2D), we first applied additional small background correction by subtracting the minimal expression value from all the data in the embryo at every time point. We also minmax normalized total CDX2 signal in each embryo using its minimal and maximal expression at the 16- and 32-cell stages (before truncating at the first 32- to 64-cell-stage division). For CDX2, we used clustering on summary data from all embryos and all relevant stages (16L, 32E and 32L) to choose a universal thresholding parameter. In particular, we applied the *GaussianMixture* method from the *sklearn.mixture* Python package to cluster the data into three clusters. Ordering the clusters by their mean expression, we annotated the bottom cluster as 0 (no CDX2 expression) and top two clusters (intermediate and high CDX2 expression) were annotated as 1, which is equivalent to using $\theta_2=\theta_3=\theta_4=0.088$.

### Discretization for SOX2
Total SOX2 signal in each nucleus was minmax normalized and background corrected analogously to the analysis on CDX2. To arrive at the summary variables $\widetilde{S}_{b,i}$ for every branch $b$, normalizing for slight branch-dependent technical differences, we took the final total SOX2 level for each stage $i$, and subtracted the total SOX2 intensity for this branch at the beginning of the 16-cell stage. Due to batch-specific differences in background SOX2 intensities, we chose a thresholding parameter that was batch dependent. In particular, for every batch, we applied the *GaussianMixture* method from the *sklearn.mixture* Python package to cluster the summary variables into two clusters. Ordering the clusters by their mean expression, we annotated the bottom cluster as 0 (SOX2−) and the top cluster as 1 (SOX2+). The batch-specific thresholding parameters varied from 0.05 to 0.17 and can be found in Fig. S2C. In this figure, for every threshold and every embryo, Gaussian log-likelihood was calculated by fitting Gaussian distributions to two subsets separated by the threshold and combining the loglikelihoods.

### Bayesian modeling
We modeled the discretized data via Bayesian networks. To this end, we applied methods from the *BayesianNetwork* class in the *pgmpy* Python package. Nodes and edges were specified as shown in Fig. 3A for the datasets with pairwise observations and as shown in Fig. 4A for the datasets with triple $(Y, S, C)$ observations. We denoted the set of all variables/nodes of the network as $V$; for a node $v \in V$, we denoted the set of its parent nodes

as $\mathcal{P}(v)$. The Bayesian network provides a decomposition of the joint probability distribution of $V$ using the *conditional probability distributions* (CPDs): $p(V) = \prod_v p(v|\mathcal{P}(v))$. Here, $p(v|\mathcal{P}(v)) = p(v)$ if a node has no parents. The nodes of the dynamic Bayesian networks are indexed by time $i$; we denoted the set of variables corresponding to the time point $i$ (time slice) by $V_i$. For the winning model M3 that we considered in this paper (Fig. 3A), for a variable $v \in V_i$, $\mathcal{P}(v) = V_{i-1}$, and the CPD corresponding to $v$ can be written in the form $p(v|\mathcal{P}(v)) = p(v|V_{i-1})$. We made use of this structure of the decomposition when generating synthetic embryos.

The candidate networks were fit to data using the maximum likelihood estimator, i.e. for a model $M$ and observed dataset $X$, $\hat{\theta} = \arg\max_\theta p(X|\theta, M)$ was used as the optimal parameter after training. To compare Bayesian models, we computed Bayesian Information Criterion (BIC) for each model via the *structure_score* method from the *pgmpy* Python package. If $n$ denotes the number of observations in the dataset $X$, and $M$ has $k$ parameters, BIC is computed via the following formula: BIC $=k \ln n - 2 \ln p(X|\hat{\theta}, M)$. Here, $\ln p(X|\hat{\theta}, M)$ is the log-likelihood of the model after training. Note that the structure score is inversely proportional to BIC, therefore higher structure scores are preferred.

We assessed the robustness of M3 predictions over multiple train-test splits in Fig. S3. For every train-test split, a random subset of 6 embryos was selected as a training set, with the rest of the embryos used for the held-out test set. With $N$=9 embryos in the YAP-CDX2 dataset, we tested all possible splits. With $N$=12 embryos in the YAP-SOX2 dataset, we used 100 random train-test splits. For each TF, after fitting M3 model to the training set, we applied variable elimination to predict the expected frequency of each induction class in the test set.

We assessed the generalization performance of the M3 model for each TF by evaluating the Expected Calibration Error (ECE; Naeini et al., 2015) on held-out test data across the same set of train-test splits, as above. For any probabilistic classification task, ECE quantifies the discrepancy between predicted posterior probabilities and empirical frequencies, measuring how well the model is calibrated. Binning the observations into groups $B_m$, $m$=1…$M$, with posterior probabilities falling into $M$ bins splitting [0, 1] interval (denoted by $i \in B_m$), ECE is defined as:

$$\text{ECE} = \sum_{m=1}^{M} \frac{1}{n} \left| \sum_{i \in B_m} z_i - \sum_{i \in B_m} \hat{p}_i \right|, \qquad (3)$$

where $n$ is the total number of observations, $z_i$ are the true test labels and $\hat{p}_i$ are the predicted probabilities of the positive label. $M$=10 was used; a lower ECE indicates better calibration.

### Generating a synthetic embryo
We generated synthetic embryos incorporating the lineage structure; in this paper, $N$=3000 embryos were used for every experiment. For every embryo, 8 lineages were sampled independently. To simulate a lineage, we made use of the CPDs of the trained model. Recall that the set of variables corresponding to the time point $i$ was denoted by $V_i$. For $i \geq 1$ and $v \in V_i$, the CPDs of the M3 model have the form of the transition matrices $p(v|V_{i-1})$; for $i$=0, the nodes have no parents and the CPDs correspond to the initial probability distributions $p(v)$, $v \in V_0$.

A simulated lineage is a collection of expression profiles of 13 synthetic cells, $x_0^a, x_1^{\{aa,ab\}}, x_2^{\{aaa,aba\}}, x_3^{\{aaaa,aaab,abaa,abab\}}$ and $x_4^{\{aaaaa,aaaba,abaaa,ababa\}}$. Here, the (redundant) subscript corresponds to the time point, and the superscript encodes the lineage structure of the tree (Fig. S3B), with the cell $x_i^s$ encoded by the binary sequence $s$ at time point $i$, giving rise to the cells $x_{i+1}^{sa}$ and $x_{i+1}^{sb}$ in case of a division and to the cell $x_{i+1}^{sa}$ otherwise. $sa$ and $sb$ denote concatenation of the sequence $s$ with $a$ and $b$, respectively; we denoted the truncated sequence $s$, with the last element removed, by $s'$. Two divisions are happening, one between time points $i$=0 and $i$=1, and the other between $i$=2 and $i$=3. The expression for the cells was independently sampled from the corresponding CPDs. More precisely, we sampled $x_0^a$ from $p(V_0)$, and sequentially sampled expression for the cells in all the subsequent time points, with expression for the cell $x_i^s$ sampled from $p(V_i|x_{i-1}^{s'})$ for $i \geq 1$. Sampling from the CPDs was realized using the *simulate* method available for Bayesian networks in *pgmpy*. Note that this strategy is appropriate for any number of variables observed over time, and only takes advantage of the

fact that the parents of the nodes in our dynamic Bayesian networks belong to the previous time slice. In particular, the same simulation strategy was applied for the pairwise and fused models.

## Data fusion via Bayesian modeling

To fuse the pairwise datasets corresponding to $(Y, S)$ and $(Y, C)$, we trained a dynamic Bayesian network on the concatenated data (Fig. 4A). Training in *pgmpy* allows for missing data. Due to the structure of the fused network, training it on both datasets results in the update of the CPDs for $Y$, i.e. $p(Y_0)$, $p(Y_i|Y_{i-1})$ for $i \geq 1$, with the CPDs for TFs, i.e. $p(G_0), p(G_i|Y_{i-1}, G_{i-1})$ for $i \geq 1$ for $G=S, C$, coinciding with the CPDs for the models only trained on paired observations. For fusion with the geometric model, the architecture from Chalifoux et al. (2025) was fused with M3 at the $Y$ chain.

## Exact inference

Bayesian networks can be used for exact inference to compute posterior distributions of the form $p(Q|E=e)$ for a set of query variables $Q$, given some observed evidence $E=e$. This task can be approached by algorithms such as variable elimination, in which all the non-query and non-evidence variables (i.e. $V \setminus \{Q \cup E\}$ where $V$ is the set of all variables in the model) are efficiently marginalized out in a pre-specified order. For exact inference, we applied the *VariableElimination* method from *pgmpy* to corresponding trained models. In particular, for variable elimination conditional on induction classes in Figs. 3E,H, we used $\{G_2=1\}$, $\{G_2=0, G_3=1\}$ and $\{G_2=0, G_3=0, G_4=1\}$ as evidence for classes $G_{16L}^+$, $G_{32E}^+$ and $G_{32L}^+$, respectively. Analogously, for Fig. 3F, $\{C_4=1\}$ and $\{Y_2=1\}$ were used as evidence; for Fig. 3I, $\{Y_2=0\}$ and $\{Y_3=1, Y_4=0\}$ were used.

## Acknowledgements
We thank the members of the Developmental Dynamics Group at the Center for Computational Biology at the Flatiron Institute for helpful discussions. We thank Lucy Reading-Ikkanda (Flatiron Institute) for assistance with the graphic design of figures and Abhishek Biswas (Princeton University) for assistance with data analysis. The Flatiron Institute is a division of the Simons Foundation.

## Competing interests
The authors declare no competing or financial interests.

## Author contributions
Conceptualization: M.A., S.Y.S., E.P.; Formal analysis: M.A., M.C.; Funding acquisition: S.Y.S., E.P.; Investigation: M.A., M.C.; Methodology: M.A., M.C.; Project administration: M.A., E.P.; Resources: B.J.; Software: M.A.; Supervision: S.Y.S., E.P.; Writing – original draft: M.A., S.Y.S., E.P.; Writing – review & editing: M.A., M.C., S.Y.S., E.P.

## Funding
Research reported in this publication was supported by the National Institutes of Health, the Eunice Kennedy Shriver National Institute of Child Health and Human Development (R01HD110577 and R01HD107026 to E.P.) and the National Institute of General Medical Sciences (R01GM134204 to S.Y.S.). The content is solely the responsibility of the authors and does not necessarily represent the official views of the National Institutes of Health. Open Access funding provided by Princeton University. Deposited in PMC for immediate release.

## Data and resource availability
The code generating the figures is available at https://github.com/MariaAvdeeva/MouseEmbryoSimulator.

## The people behind the papers
This article has an associated 'The people behind the papers' interview with some of the authors.

## Peer review history
The peer review history is available online at https://journals.biologists.com/dev/lookup/doi/10.1242/dev.204717.reviewer-comments.pdf

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
