## [Peer Review File · Development (Cambridge, England)]

Generative model for the first cell fate bifurcation in mammalian development

Maria Avdeeva, Madeleine Chalifoux, Bradley Joyce, Stanislav Shvartsman and Eszter Posfai

DOI: 10.1242/dev.204717

Editor: Paul Francois

Review timeline

Original submission: 11 February 2025

Editorial decision: 17 March 2025

First revision received: 20 June 2025

Accepted: 17 July 2025

Original submission

First decision letter

MS ID#: dev.204717

MS Title: Generative model for the first cell fate bifurcation in mammalian development

Authors: Maria Avdeeva; Madeleine Chalifoux; Bradley Joyce; Stanislav Shvartsman; Eszter Posfai

Article Type: Research Article

Dear Dr Posfai,

I have now received all the referees reports on the above manuscript, and have reached a decision. The referees' comments are appended below, or you can access them online: please go to .

The overall evaluation is positive and we would like to publish a revised manuscript in Development, provided that the referees' comments can be satisfactorily addressed. Please attend to all of the reviewers' comments in your revised manuscript and detail them in your point-by-point response. If you do not agree with any of their criticisms or suggestions explain clearly why this is so. If it would be helpful, you are welcome to contact us to discuss your revision in greater detail. Please send us a point-by-point response indicating your plans for addressing the referees' comments, and we will look over this and provide further guidance.

Reviewer 1

SUMMARY OF THE ADVANCE MADE IN THIS PAPER AND ITS POTENTIAL SIGNIFICANCE TO THE FIELD

In this study Avdeeva et al. build a generative model of the TE vs ICM cell fate bifurcation in the early mouse embryo based on their live-image datasets. This integrates time-dependent statistics of the cell fate specification, specifically, changes in Yap, Cdx2 and Sox2 expression. They built two independent models to show that Yap dynamics are necessary or sufficient for Cdx2 and Sox2 expression. Furthermore, they fused the two models and predicted the appearance of Cdx2/Sox2 double-positive cells, which is confirmed by their experimental data.

Although the use of dynamic Bayesian networks to model gene expression is not new, applying this framework in developing embryos is a simple, effective, and elegant new approach. Given the

novelty and the power of quantitative analysis presented in this study and the potential applicability to other developmental systems, this study would be suitable for publication in Development after addressing the following issues.

SUGGESTIONS TO AUTHORS

1. While the analysis is performed in high quality, the number of samples is limited to N=6 embryos for Yap/Sox2/H2B, N=9 for Yap/Cdx2/H2B, and N=2 for Yap/Cdx2/Sox2/H2B live-image datasets. Could the authors increase the sample size, and perform statistical analyses to compare the model predictions with the experimental observations, including variabilities, potentially offering stronger and more significant conclusion?
2. As their image datasets should have the information of cell/nuclear position, could the authors incorporate it (inside or outside position) into their model, e.g. as an upstream regulator for Yap? Could this increase the precision of the model and its predictions, and possibly show the origin of Cdx2/Sox2 double-positive cells, as described in Discussion?
3. The main text is often difficult to follow, due to the extensive use of technical words and concepts. Could the authors modify the description of their model and its analysis, so that it is more easily accessible to readers not familiar with Bayesian statistics?

Reviewer 2

SUMMARY OF THE ADVANCE MADE IN THIS PAPER AND ITS POTENTIAL SIGNIFICANCE TO THE FIELD

The authors generate a Bayesian model for two lineage-marking transcription factors, SOX2 and CDX2, and one of their regulatory effectors, YAP, based on live imaging of early mouse embryos. The selected Bayesian model allows investigation of the most likely genetic circuits involved. It shows that there is a delay with which Yap affects CDX2 and SOX2 expression, that YAP expression is a necessary condition for CDX2 expression and YAP loss a necessary but not sufficient condition for SOX2 expression. The authors conclude that another pathway, such as Notch2, may be required to determine downstream expression for CDX2 and to understand SOX2 variability. The paper is clear, accessibly written, and both approach and conclusions are interesting to the field; the Bayesian and binarized approach to interpret dynamics should be of broad relevance. I just have minor comments.

SUGGESTIONS TO AUTHORS

1. Could the authors comment on how much the threshold for the binarization affects the model choice? Would this affect the choice of the model or predominantly individual values of the transition matrix?
2. I was surprised that the model M4 was worse than M3, as M3 is contained in M4. Could the authors explain why this is? Further, could the authors comment on a model where Yap has no longer an effect in the later stages 32E and 32L is also reasonable, to investigate whether late expression of Yap still has an impact on the lineage?
3. The comparison of generated with experimental dynamics in Fig 4 is very nice, as are the interesting fluctuations in Sox2. I was surprised by the discretization in Fig4D, as it seems that the threshold for Sox2 expression used to obtain the true discretized results is rather low (ca 0.2). Do the authors have a sense for what molecular concentrations this fluorescence could correspond to? What pathways could be involved in Sox2 expression and explain its variability?

First revision

Author response to reviewers' comments

We thank the Reviewers for their suggestions. In the revised version we made every effort to address the questions and implemented essentially all the suggested changes. Key revisions include:

1. **Robustness to coarse-graining parameters.** We have systematically examined the effect of threshold selection during the data coarse-graining step on the choice of the top-performing model and concluded that this choice is robust to the thresholding parameter (Supp. Fig. 3C).
2. **Expanded experimental dataset and improved validation.** We have doubled the number of embryos live imaged and analyzed for YAP and SOX2 and have done extensive model training/validation for both YAP-SOX2 and YAP-CDX2 models. Supp. Fig. 3D supports the robustness of our predictions and Supp. Fig. 3E shows evidence that our probabilistic models generalize well to unseen data.
3. **Data fusion extension.** We have extended our data fusion approach to incorporate information about the relative exposed surface areas of the blastomeres (Supp. Fig. 4C).
4. **Improved clarity.** Wherever possible, we have simplified the technical presentation of our approach.

The responses to each question are included below.

Reviewer 1: SUMMARY OF THE ADVANCE MADE IN THIS PAPER AND ITS POTENTIAL SIGNIFICANCE TO THE FIELD

In this study Avdeeva et al. build a generative model of the TE vs ICM cell fate bifurcation in the early mouse embryo based on their live-image datasets. This integrates time-dependent statistics of the cell fate specification, specifically, changes in Yap, Cdx2 and Sox2 expression. They built two independent models to show that Yap dynamics are necessary or sufficient for Cdx2 and Sox2 expression. Furthermore, they fused the two models and predicted the appearance of Cdx2/Sox2 double-positive cells, which is confirmed by their experimental data.

Although the use of dynamic Bayesian networks to model gene expression is not new, applying this framework in developing embryos is a simple, effective, and elegant new approach. Given the novelty and the power of quantitative analysis presented in this study and the potential applicability to other developmental systems, this study would be suitable for publication in Development after addressing the following issues.

SUGGESTIONS TO AUTHORS

1. While the analysis is performed in high quality, the number of samples is limited to N=6 embryos for Yap/Sox2/H2B, N=9 for Yap/Cdx2/H2B, and N=2 for Yap/Cdx2/Sox2/H2B live-image datasets. Could the authors increase the sample size, and perform statistical analyses to compare the model predictions with the experimental observations, including variabilities, potentially offering stronger and more significant conclusion?

We thank the Reviewer for this valuable suggestion. Due to the smaller number of Yap/Sox2/H2B embryos in the original submission, we focused our experimental efforts on expanding this dataset. We have collected 6 more embryos for Yap/Sox2/H2B and included them into our analysis resulting in N=12 for the YAP-SOX2 dataset. The figures and text have been updated accordingly. We have also performed additional statistical analyses to validate our predictions for both YAP-SOX2 and YAP-CDX2 datasets.

To compare model predictions with the experimental observations, we used many training-testing splits in which 6 embryos were used as the training set and the rest (3 embryos for every split for YAP-CDX2, and 6 embryos for YAP-SOX2) were used as the held-out test set. For every split, after training our model we made predictions of the TF induction class frequencies on the test set. We first demonstrated that our mean frequency predictions on the test set are robust with respect to the train-test split (Supp. Figs. 3D). To validate our posterior predictive distribution (and not just the mean predicted frequencies) even further, we compared our predicted class probabilities to the true observed classes on each trajectory in the test set using expected calibration error metric (see Methods for details). This metric is appropriate for probabilistic classifiers and we observed good calibration with average ECEs < 0.1 (Supp. Fig. 3E). These results indicate reliable predictions of induction class probability for any new independent observation.

To illustrate the results for a random train-test split of the YAP-SOX2 data, below we compare class frequencies predicted after training (bar: mean frequencies, errorbar: 1 std, N=3000 simulated embryos) with the corresponding distributions in the test set (N=6 embryos).

For both means and standard deviations (to test variability), we compared the distribution of the statistic predicted by the trained model (over 500 groups of 6 embryos, shown in blue) with the observed value in the test set (6 embryos, black dashed lines). We found that the observed values lie within 95% CI (blue dashed lines) for all classes.

2. As their image datasets should have the information of cell/nuclear position, could the authors incorporate it (inside or outside position) into their model, e.g. as an upstream regulator for Yap? Could this increase the precision of the model and its predictions, and possibly show the origin of Cdx2/Sox2 double-positive cells, as described in Discussion?

We agree that this is an exciting direction for new analysis. In a recent paper (Chalifoux et al 2025, PMID: 40060487), we have applied an analogous Bayesian network approach to a dataset where YAP was observed alongside H2B and a membrane reporter. In that model, cell position was measured

via its relative exposed area (denoted by R below) which is defined as the proportion of its surface area that is not in contact with the other cells of the embryo. 3 possible positions were defined based on R at every stage, inner ($R=0$), intermediate ($R=1$), and outer ($R=2$). Due to the similarity in the approach, we could fuse the RY network to the YCS network introduced in this paper (see the architecture on the left below). Since position is upstream of YAP in this network and we only trained this network on RY and YCS data, this should not increase the precision of our predictions except via the increase in the number of Y observations. On the other hand, with this approach one can indeed predict position dynamics of the double positive cells (and other classes). In particular, most of the double positive cells are predicted to assume the inner position by the 32E stage and retain it at the 32L stage. These predictions (on the right below) are now included in Supp. Fig. 4C.

3. The main text is often difficult to follow, due to the extensive use of technical words and concepts. Could the authors modify the description of their model and its analysis, so that it is more easily accessible to readers not familiar with Bayesian statistics?

We have now modified relevant sections, most extensively, the section “Bayesian modeling of pairwise dynamics”. While we cannot avoid most of the technical details because of our focus on the computational approach, we added some clarifications and simplified the language, e.g., removed technical terms like “Markov chain” and avoided references to conditional probability unless it was necessary.

Reviewer 2: SUMMARY OF THE ADVANCE MADE IN THIS PAPER AND ITS POTENTIAL SIGNIFICANCE TO THE FIELD

The authors generate a Bayesian model for two lineage-marking transcription factors, $SOX2$ and $CDX2$, and one of their regulatory effectors, YAP , based on live imaging of early mouse embryos. The selected Bayesian model allows investigation of the most likely genetic circuits involved. It shows that there is a delay with which Yap affects $CDX2$ and $SOX2$ expression, that YAP expression is a necessary condition for $CDX2$ expression and YAP loss a necessary but not sufficient condition for $SOX2$ expression. The authors conclude that another pathway, such as $Notch2$, may be required to determine downstream expression for $CDX2$ and to understand $SOX2$ variability. The paper is clear, accessibly written, and both approach and conclusions are interesting to the field; the Bayesian and binarized approach to interpret dynamics should be of broad relevance. I just have minor comments.

SUGGESTIONS TO AUTHORS

1. Could the authors comment on how much the threshold for the binarization affects the model choice? Would this affect the choice of the model or predominantly individual values of the transition matrix?

We thank the Reviewer for this suggestion. We have now conducted robustness analyses to see how the threshold affects the model choice.

While the YAP-CDX2 dataset and analysis stayed the same, we have now collected 6 more embryos for Yap/Sox2/H2B and included them into our analysis resulting in N=12 for the YAP-SOX2 dataset. It is worth mentioning that we have observed batch-specific differences in the background SOX2 intensities between embryos. As a result, we have modified our thresholding procedure for SOX2 making the thresholds batch-specific. Details are now included in the Methods and in Supp. Figs. 2C,E.

For both datasets, we analyzed a grid of 101 threshold values θ between 0 and 1 and compared the structure score (on the training set of 6 embryos each) for the 4 models of interest for every θ . Note that we used a universal threshold in our robustness analysis for both YAP-CDX2 and YAP-SOX2 data. As you can see below (Supp. Fig. 3C), for YAP-CDX2 data, the winning model is M3 for θ between 0.05 and 0.49 and for YAP-SOX2 data, for θ between 0.0 and 0.73 except $\theta = 0.05$. For YAP-CDX2, the optimal threshold used in the paper is shown, for YAP-SOX2 optimal values are batch-specific, and vary between 0.05 and 0.17 (see Supp. Fig. 2C). From this analysis, we concluded that M3 outperforms other models over the relevant range of thresholding parameters.

2. I was surprised that the model M4 was worse than M3, as M3 is contained in M4. Could the authors explain why this is?

The reason for this is in the scoring metric that we chose to apply to each model. Indeed, for model selection we use Bayesian Information Criterion which penalizes the models for the number of parameters: $BIC = k \ln(n) - 2 \ln(L)$ where L is the likelihood for the optimal set of parameters, n is the number of observations, k is the number of parameters (see Methods). Lower BIC models are preferred. The structure score reported in Supp. Fig. 3A is $-\frac{1}{2} * BIC$ leading to models with higher structure scores being preferred.

M4, indeed, contains more edges than M3 (with the edges of M3 being the subset of the edges of M4) resulting in higher likelihood for M4. However, the number of parameters for M4 is also larger, therefore, both lower and higher BICs could be expected for M3 when compared to M4. For our data, $BIC(M3) < BIC(M4)$ so we selected M3.

Further, could the authors comment on a model where Yap has no longer an effect in the later stages 32E and 32L is also reasonable, to investigate whether late expression of Yap still has an impact on the lineage?

This is an interesting suggestion. To analyze this, we have now included M5: a model like M3 but with Y3 -> G4 edge removed. This network eliminates the effect of YAP on the downstream gene expression at the 32-cell stage. The structures of the networks are shown on the left. The training structure scores for all the networks including M5 are shown on the right. We found that, for both YAP-CDX2 and YAP-SOX2, M3 outperforms M5 as well as other models. Interestingly, for YAP-CDX2, model performance for M5 is very similar to M3. This suggests that YAP effects are still relevant for SOX2 at the 32 cell stage, while CDX2 expression is mostly set by 16 cell stage YAP levels.

Vars	YC	YS
M1	-600.95	-598.58
M2	-498.66	-543.43
M3	-479.66	-523.41
M4	-509.17	-559.12
M5	-480.22	-535.71

3. The comparison of generated with experimental dynamics in Fig 4 is very nice, as are the interesting fluctuations in Sox2. I was surprised by the discretization in Fig4D, as it seems that the threshold for Sox2 expression used to obtain the true discretized results is rather low (ca 0.2). Do the authors have a sense for what molecular concentrations this fluorescence could correspond to? What pathways could be involved in Sox2 expression and explain its variability?

Unfortunately, we do not have a sense for what molecular concentration the measured fluorescent intensity corresponds to, as this would require a different, time-consuming and technically challenging, experimental approach (e.g. stepwise photobleaching or fluorescence correlation spectroscopy). However, we suspect that Sox2 falls into binary “not expressed” or “expressed” categories, and therefore a low threshold to distinguish the two is not that surprising. Furthermore, our sample-by-sample thresholding analysis (Supp. Fig. 2C) demonstrated that the optimal threshold for Gaussian loglikelihood is low consistently between samples.

The upstream regulators of Sox2 expression are key unanswered questions, which we mention in the Discussion section (“These observations highlight that we are lacking critical regulators of SOX2 in the embryo, either an additional repressor or an activator, whose activity is highly heterogeneous among ICM cells.”). We have added an additional sentence to the Discussion: “Notably, factors such as OTX2 and CARM1 have been suggested to regulate Sox2 expression in the embryo (Acampora et al 2016, PMID: 27292645; Torres-Padilla et al 2007, PMID: 17215844). Whether endogenous heterogeneities in these factors can explain SOX2 variabilities will need to be investigated in the future.”

Second decision letter

MS ID#: dev.204717R1

MS Title: Generative model for the first cell fate bifurcation in mammalian development

Authors: Maria Avdeeva; Madeleine Chalifoux; Bradley Joyce; Stanislav Shvartsman; Eszter Posfai
Article Type: Research Article

Dear Dr Posfai,

I am happy to tell you that your manuscript has been accepted for publication in Development, pending our standard publication integrity checks.

Reviewer 1

In the revised manuscript the authors addressed the issues raised by the reviewer and the study is suitable for publication.

Reviewer 2

SUMMARY OF THE ADVANCE MADE IN THIS PAPER AND ITS POTENTIAL SIGNIFICANCE TO THE FIELD

The authors combine live imaging and Bayesian generative modeling in order to better understand early cell fate development and decision making in mouse embryonic development. To do so, they measure established transcription factors dynamically, at 5 interesting stages, up until the 32-cell stage of the embryo. The authors then use this data to select a dynamic Bayesian network that is consistent with the trajectories they observe. Exploring this network yields an interesting understanding of the transcription factor network, e.g. that YAP is necessary for CDX2 expression but cannot predict the onsite timing, and necessary but not sufficient for Sox2 expression.

I found the paper interesting and enjoyable to read - I think that the presentation improved compared to the previous version, and was impressed by the additional dataset and the consistency of previous results also with this dataset. The authors answered all my questions clearly, and I appreciated the additional discussion of the threshold value in their answers and also in the supplement of the paper.

I am more than satisfied with the authors comments, and think that this paper is should be accepted for publication.

SUGGESTIONS TO AUTHORS

On the proof stage, it might be good to check Ref 47. Upon re-reading, I also thought that perhaps there is a way to add another sentence in the introduction or the abstract to target the paper to modelling/theory researchers interested in trajectories, which seem to be enjoying a growing interest from both the non-linear dynamics and information communities at the moment (perhaps even just having the word "trajectories" in the introduction). However, this is by no means necessary, and the authors will know what is best.